# RelScene: A Benchmark and baseline for Spatial Relations in text-driven 3D Scene Generation

## ABSTRACT

Text-driven 3D indoor scene generation aims to automatically generate and arrange the objects, which form a 3D scene that accurately captures the semantics detailed in the given text description. Recent works have shown the potential to generate 3D scenes guided by specific object categories and room layouts but lack a robust mechanism to maintain consistent spatial relationships in alignment with the provided text description during the 3D scene generation. Besides, the annotations of the object and relationships of the 3D scenes are usually time- and cost-consuming, which are not easily obtained for the model training. Thus, in this paper, we conduct a **dataset** and benchmark for assessing spatial relations in text-driven 3D scene generation, which contains a comprehensive collection of 3D scenes, including textual descriptions, annotating object spatial relations, and providing both template and free-form natural language descriptions. We also provide a **pseudo description feature generation method** to address the 3D scenes without language annotations. We design an aligned latent space for spatial relation in 3D scenes and text description, in which we can sample the features according to the spatial relation for the few-shot learning. We also propose **new metrics** to investigate the ability of the approach to generate correct spatial relationships among objects.

## KEYWORDS

3D scene generation, 3D Vision-Language Learning, Dataset and Evaluation Metric, Few shot learning

## 1 INTRODUCTION

Text-driven 3D indoor scene generation is an important task to create realistic rooms with suitable objects(i.e., furniture) and layouts. It requires the ability of the model to generate scenes automatically and maintain consistent semantics in alignment with the provided text description. It has gained much attention for its potential applications in interior design, virtual reality, and video games[26]. It can significantly minimize repetition and boost productivity by eliminating manual scene creation requirements. Moreover, it provides interior designers with a valuable resource to quickly generate room designs and gather client feedback, opening up numerous practical applications.

Recent works have shown the potential to generate 3D scenes guided by specific object categories and room layouts. They have

*ACM MM, 2024, Melbourne, Australia*
© 2024 Copyright held by the owner/author(s). Publication rights licensed to ACM.
ACM ISBN 978-x-xxxx-xxxx-x/YY/MM
https://doi.org/10.1145/nnnnnnn.nnnnnnn

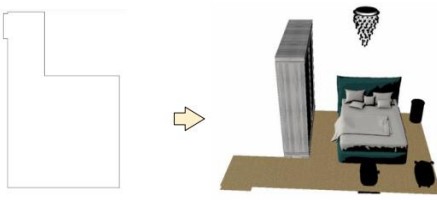

(a) No Semantic Constraint

There is a bed and a some wardrobes in the room.

(b) Instance-level semantic constraint

Nightstand is **beside to** single bed. Wardrobe is **left** to single bed. Chair is **near** the wardrobe

(c)Instance and spatial relation semantic constraints

**Figure 1: Different annotations of the 3D scene generation.**

achieved tasks like automatic layout synthesis, scene completion, and object suggestion[10, 13, 21]. For example, Sync2Gen [23] utilizes a trained parametric prior distribution to effectively control the generation of unrealistic indoor scenes by feed-forward neural models. ATISS [13] utilizes autoregressive transformers to predict object locations in sequence. However, they are not designed to maintain consistent spatial relationships with the description. Furthermore, due to the time- and cost-consuming of language annotations, language annotations of the spatial relationships in the 3D scene are not easily obtained for the training. There is also no available benchmark to evaluate the abilities of the approach to control the spatial relationships for text-driven 3D indoor scene generation precisely.

To tackle these issues, we propose a new benchmark for assessing spatial relationships in text-driven 3D scene generation. We offer the dataset and spatial relations metrics to assist research in the text-driven 3D scene generation domain. As shown in Fig. 1, compared with the current works to generate 3D scenes without semantic constraints (a) or only with instance-level semantic constraint (b), our new proposed dataset and method aims to address

both instance and spatial relation semantic constraints (c) for the 3D scene generation. The task requires combining the objects with the proper layout and understanding and effectively controlling the relationship among multiple objects during the generation process. We also investigate text-driven 3D scene generation tasks with few language annotations. The main contribution can be summarized as follows:

(1) **New dataset.** We propose a new dataset for text-driven 3D scene generation. Currently, there is no suitable dataset with text description of the scene for text-driven 3D scene generation. We extend the 3D-FRONT to construct a text-driven 3D scene generation dataset with spatial relations description. We annotate the spatial relations of the objects in the scenes and provide two types of text descriptions, including the template descriptions and free-form nature language descriptions for the text-driven 3D scene generation dataset.

(2) **Pseudo description feature generation method** for few-shot text-driven 3D scene generation. We design an aligned latent space for spatial relation in 3D scenes and text description, in which we can sample the features according to the spatial relation for the few-shot learning.

(3) **New metrics.** We propose two metrics to evaluate the accuracy of the described spatial relationship generated in the 3D scene. We evaluate the spatial relationship of the scenes from three aspects: local and relation metrics. The local metric evaluates the accuracy of the pair relation between two objects, while the relation metric evaluates the model's ability to address different relation types.

We conduct the experiments on our new proposed dataset and compare our benchmark method with the current work on both the newly proposed metrics and the traditional metrics. The results show that our benchmark method showcases its superior flexibility in scene qualities and spatial relations accuracy.

The remainder of this paper is organized as follows. Section 2 briefly surveys the related works. Section 4 introduces the benchmark method we propose to address the text-driven 3D scene generation. Section 3 presents the approach and details we design to annotate and extend the 3D-FRONT datasets for text-driven 3D scene generation. In Section 5, we proposed new metrics to evaluate how accurately the described spatial relationship is generated in the 3D scene. Then, the experiments are introduced in Section 6 to compare the benchmark method and current work in our proposed dataset. Finally, Section 7 concludes this paper.

## 2 RELATED WORK

**3D scene datasets with Language**: Similar to our work, prior works have also explored grounding language in 3D. Notably, Chang et al.[3] model spatial knowledge by leveraging statistics in 3D scenes. They created a dataset with 609 annotations between 131 object pairs in 17 scenes for spatial relations. Also, Chang et al. created a model for generating 3D scenes from text and created a dataset of 1129 scenes from 60 seed sentences. Concurrent with our work, Panos et al.[1] proposed ReferIt3D, a benchmark for contrasting objects in 3D using natural and synthetic language.

However, unlike in prior works, our dataset provides the spatial relations of the objects in the scenes with two types of text descriptions, including the template descriptions and nature language descriptions for the text-driven 3D scene generation dataset. Our dataset contains more than 6000 scenes, which is much larger than the datasets mentioned above.

**3D Indoor scene generation**: Different from the 3D Scene Reconstruction [8], 3D indoor scene generation aims to explore object layout generation. Current works usual setting of task in indoor scene synthesis is to retrieve 3D models from a given database and predict the model positions for the generation of semantically and functionally realistic indoor scenes.

Graph-based Scene Synthesis represents scenes as graphs, which has been extensively studied in the last years. 3D scenes can be hierarchically decomposed into multiple semantic levels of content. Inspired by the works of indoor scene parsing [5, 22, 28] and 3D scene understanding [25, 29], the researchers encode the 3D scenes into different forms, such as parse trees[14], adjacency matrices[30], scene graphs[11] and scene hierarchies[9], and then synthesize 3D scenes by decoding those form. Huang et al. [7] propose to apply holistic scene grammar to parse scenes as hierarchical structures for reconstruction from a single image. Armeni et al.[2] consider representing the entire building with rooms and objects in a scene hierarchical representation. More recently, the performance has been significantly enhanced by leveraging deep learning-based approaches. Wang et al. [19] introduced an image-based generative model with relation graphs. Li et al. [9] leveraged a recursive neural network to model the furniture within a room with four walls representing the objects and their relationships in a hierarchy, and employed recursive neural networks.

Some works also present a framework for interior scene synthesis with spatial prior neural networks. Ritchie et al.[15] and Wang et al.[20] proposed image-based deep convolutional generative models for related purposes. Zhang et al.[30] tackled the challenge of free-form generation without imposing floor constraints using a generative adversarial network and a hybrid representation. Due to the effectiveness of transformers, the autoregressive-based model of scene synthesis attempts to model the scenes. SceneFormer [21] accomplished faster and more realistic 3D scene generation by leveraging the self-attention of transformers. The method predicts object locations in sequence based on either room layout or text descriptions. ATISS [13] also utilizes autoregressive transformers for tasks like automatic layout synthesis, scene completion, and object suggestion. To further advance location recommendations within incomplete indoor scenes, Zhou et al.[31] adopted neural message passing. The method enables learning spatial and structural relationships between the objects by predicting the probability of newly added objects.

**Text-Conditioned Scene Synthesis**: There are also other works learning to produce furniture layouts under language, activity, human, and action constraints. Text-conditional generation or text-to-scene translation tasks have been studied in recent years. Current text-conditioned 3D scene synthesis relies on retrieving a similar scene from the database, which does not enable generating scenes according to the text description. For example, Ma et al.[12] introduce a natural language framework to edit 3D indoor scenes with an annotated large 3D scene database. They parse the command

of edition from the users and transform it into a semantic scene graph to retrieve the corresponding sub-scenes from the databases that match the command. The 3D scene is synthesized by aligning the augmented sub-scene with the user's current scene. Chang et al. [4] propose an interactive text-to-3D scene generation system, which allows users to provide text as input. The system can use the spatial knowledge learned from the existing databases to infer the layout of the objects and form a scene that matches the input description. Zhang et al. [27] propose an interactive scene synthesis tool to quickly picture various potential synthesis results by simultaneously editing groups of objects.

In this paper, our proposed work can directly generate the layout of the objects under the constraints of the text description, which is more flexible and accurate for implementing specific design ideas for building 3D indoor scenes. We design a few-shot method to apply the scenes without text annotation to train the model. We also provide a new metric to investigate the ability of approach models to generate correct spatial relationships among objects.

## 3 NEW DATASET AND METRICS FOR TEXT-DRIVEN 3D SCENE GENERATION

While existing research has introduced some datasets containing 3D scenes, these datasets were not specifically tailored for text-driven 3D scene generation, making it challenging to assess the precision of the generated outcomes. To address the deficiency of suitable datasets for constructing 3D scene models from textual descriptions, we introduce a new dataset built upon the foundation of the 3D-FRONT dataset. Our new proposed dataset has scenes with multiple objects and textual descriptions for the generation task, which includes 13 types of relationships of the objects. (More details can be found in the Supplementary materials)

### 3.1 Scene Data Processing and Text generation

**Scene Data Processing.** Our dataset is extended from 3D-FRONT dataset[6]. We select samples from the original dataset in three categories: bedrooms, living rooms, and dining rooms. In accordance with the methodologies outlined in recent studies[17][10], we have applied the identical dataset filtering procedures as employed in the ATISS framework. These procedures involved excluding scenes that exhibited excessive complexity, excessive simplicity, and the absence of typical object relationships. This process yielded 4041, 900, and 813 scenes in their respective subsets. In all these scenes, the number of objects ranged from 3 to 13, ensuring that the quantity of textual descriptions would not be excessive or insufficient. Each scene contains several types of information, including the object class, object positions, orientations, and object coordinates marked with eight points. We extract this information to generate template-based textual descriptions and provide essential data for natural language models.

**Template-based textual descriptions generation.** The original 3D-FRONT dataset has no textual description for text-driven 3D scene generation. To equip the 3D-Front dataset with text descriptions, we propose an algorithm for labeling fundamental spatial relationships by drawing insights from the extracted scene data. The initial text content contains $C_n^2$ (n is the number of objects in the scene) textual descriptions and is redundant. Hence, we applied

a filtering algorithm based on probability to handle the previously acquired set of text descriptions. The filtering model considers the occurrences of typical relationships, volumes, and distances between objects to evaluate the importance of specific texts. By employing this recursive calculation approach, we establish a mechanism that prioritizes selecting significant relationships and those involving the key objects (the objects with big volumes and more relationships) within the scene. Simultaneously, it maintains a certain likelihood of selecting less conspicuous relationships, thereby upholding the diversity of textual descriptions.

**Natural language descriptions generation.** To get closer to the text-driven 3D scene generation application scenario, we generate natural language descriptions with ChatGPT, a powerful language model for dialogue. The rewritten description by the ChatGPT is more natural to the human inputs. It provides a more robust evaluation of the text-driven 3D scene generation models' capacity to extract and comprehend crucial information from natural language. Additionally, these descriptions offer a broader range of challenges and diversity for the task.

## 4 APPROACH

In the text-driven 3D scene generation process, the input is a textual description, and the task involves creating 3D indoor scenes while adhering to specified spatial relationship constraints described in the text. Building on the principles of contemporary 3D indoor synthesis approaches, additional constraints, such as the floor shape (represented as a top-down orthographic projection of the floor), are also considered. An overview of our approach is illustrated in Figure2. The approach is designed in an auto-regressive process that inputs a top-down orthographic projection of the floor, an incomplete scene, and the accompanying text description. According to the inputs above, the approach generates the attributes of the objects, which normally contain 3D coordinates, rotation, and size, to form the final scene.

As shown in the left part of Fig.2, Our proposed approach initially employs a transformer-based scene encoder model to handle diverse inputs, including layout features and the objects within the scene. The outputs of the scene encoder are then applied to an attribute prediction network to decide the attributes of the newly generated object. During the generation process, the text features are applied to control the relationships of the objects through the cross-attention mechanism.

Due to the text description of the 3D scenes is not easy to obtain in the application, we also consider the few-shot setting of the text-driven 3D scene generation, while most of the scenes for the model training lack text descriptions. Beyond the directly utilized text description of the scene, as shown in the right part of Fig.2, we propose a pseudo feature generation method to utilize the scenes without description, which can further improve the abilities of our approach under the few-shot setting.

### 4.1 Scene Feature encoders

Our approach receives various inputs, encompassing the floor layout, objects in the partially assembled scene, and the accompanying text description. To comprehensively incorporate all the information necessary for generating new objects, we introduce multiple

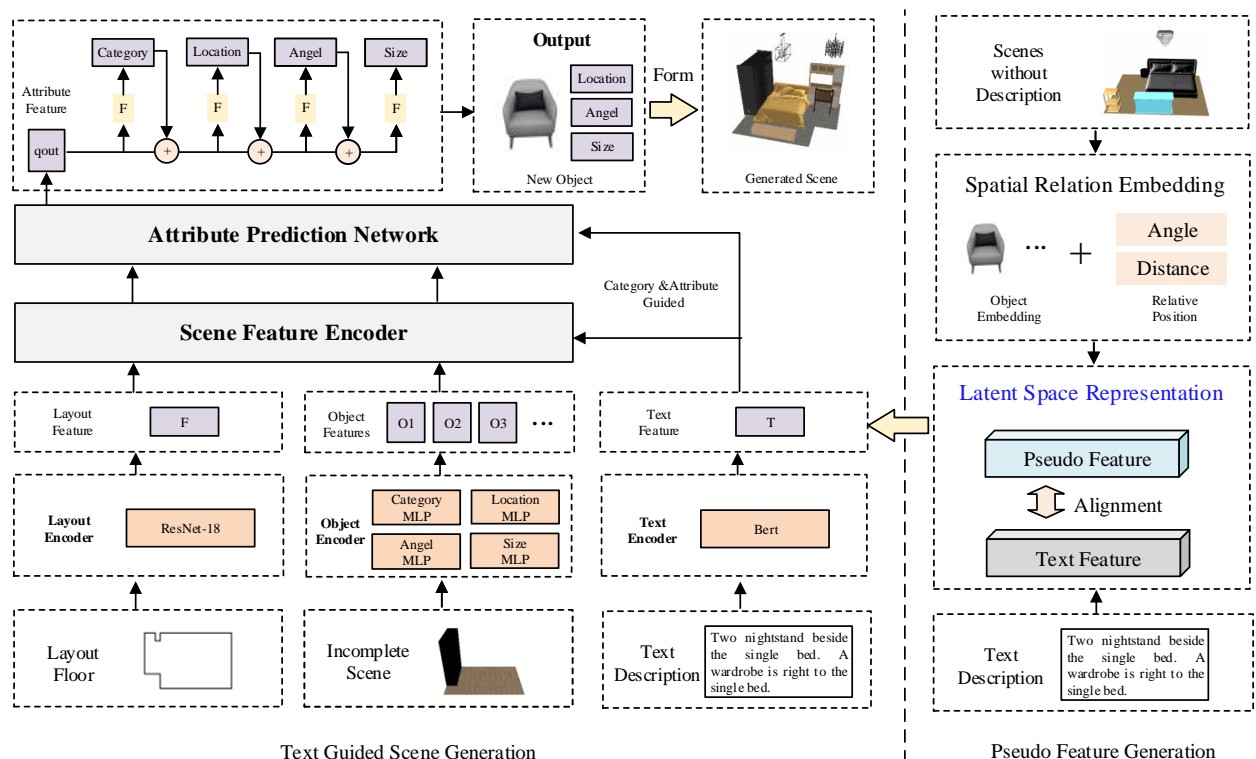

**Figure 2: The framework of our proposed method.**

feature encoders, including layout, object, and text encoders, as a vital component in the initial phase of our approach.

The role of the layout encoder is to encode the layout's shape into a feature vector denoted as $F$. This is accomplished using the ResNet-18 network, which extracts features from a top-down orthographic projection layout image. On the other hand, the object encoder is designed to encode various attributes of the objects generated in the early stages, including their category, location, angles, and size. We transform the $Fc \in R^C$, the size $Fs \in R^3$, the location of the centroid $Fl \in R^3$, and the angles $Fa \in R^1$ of the objects into representations and concatenate them into a single vector. Specifically, the category is embedded using a fully connected layer, while positional encoding[18] is applied to the remaining attributes. Besides, The text encoder is a pre-trained language model to encode the text descriptions into the language model space denoted as $T$.

Subsequently, we implement a transformer model to address the layout feature $F$, the object feature $O$, and text description features $T$ for text-driven 3D scene generation. We first concatenate the layout features $F$ and object features $O$ into a sequence to feed into the transformer model.

## 4.2 Attribute prediction network

The attribute prediction network plays a crucial role in acquiring the ability to generate objects with appropriate compositions. As

illustrated in Fig.2, this network is purposefully designed to forecast the attributes of the next object in the scene.

Firstly, the network uses a transformer model to address the output of the token by the scene feature encoders. Specifically, there is an empty token that represents a new object for the generation process. The output of this empty token is regarded as the attribute features.

Initially, it predicts the object's category and then concatenates the embedded location features $F_c$ with the attribute feature, resulting in $P \times F_c$, to predict the object's angle, Where $P$ is the injected features mentioned in the Eq.2. Similarly, for object size prediction, the location, angle, and size are predicted, embedded, and combined with the attribute feature in an autoregressive manner.

## 4.3 Cross-attention semantic injection

To apply the text description to guide the 3D scene generation, we designed a two-stage cross-attention semantic injection for the generation model in both the scene encoder stage and the attribute prediction stage.

Both the scenes feature encoders and the attribute prediction network takes a transformer to model the different inputs. Thus, we design a cross-attention mechanism to inject the semantics of

the description for generation, which is described by the equations:

$$Attention(Q, K, V) = softmax(\frac{QK^T}{\sqrt{d}})V \qquad (1)$$

where $Q = f_q(q)$ is obtained from the latent features $q$ of the transformer encoder, which is further input into the attribute prediction network. $K = f_k(c)$, and $V = f_v(c)$ is obtained from the embedding of the text description $c$. $f_q, f_k, f_v$ are mapping function. Finally, We apply a residual way to update the latent features $q$ as follows:

$$P = q + f_p(Attention(Q, K, V)) \qquad (2)$$

where $f_p$ is the mapping function.

### 4.4 Pesudo Feature generation for few-shot learning

Due to the lack of text description of the 3D scenes in the application, how to utilize the scenes without text description to train a text-driven 3D scene generation model is necessary to be studied for real-world application. Beyond the directly utilized text description of the scene, a logical approach to training the model using automatically generating text descriptions. Actually, we do not need to generate real text descriptions during the training stage. A more efficient way is to develop a latent space of spatial relations for the scenes. During the training stage, we can sample the pseudo features from the latent space according to scenes, while in the test stage, we can map the text description into the latent space for generation.

Instead of relying on directly generated text descriptions, we propose pseudo-feature generation for spatial relationships within 3D indoor scenes during training. First, we apply the few-shot samples to learn a latent space of the spatial relation.

Given a 3D indoor scene, we directly embed the spatial relation of the two objects by concatenating the object embedding and the positions. For the objects in the 3D scene, we embed the object into the feature $O$, which contains the category, location, angle, and size features of the objects from the feature encoder. Thus, we can randomly choose two objects in the scene and concatenate the object features $O_i$ and $O_j$ into $S = O_i \times O_j$ as the scene representation of the objects and their relation in the scene. The representation above cannot apply to the training because it does not correlate with the semantic space of the text description. Thus, the next step is to learn a common space to bridge the scene representation and the text description.

We utilize the few-shot samples to help build the common space to achieve the above purpose. We adopt a pre-trained language model to encode the text description of the two objects into the text features $T$. Then, two mapping functions $F_S$ and $F_T$ are designed to reduce the gap between the text features $T$ and the scene representation $S$. We apply the few-shot samples to pre-train the mapping functions by minus the difference between matched mapping features $F_S(S)$ and $F_T(T)$ in the learned latent space.

In this way, we construct the relation between the scene representation and the text description. It allows us to apply the scene representation to replace the text description for training and apply the scene without text description to train the text-driven generation model. Through the learned common space, we can utilize the mapping scene representation $F_S(S)$ as the pseudo feature to

replace the text description in the training stage. While in the testing stage, we use the mapping text description features $F_T(T)$ to control the generation process by input text description.

### 4.5 Training and the Objectives

This section summarizes the training objectives of our approach in the training stage.

During the learning of the latent space of spatial relations, we learn two mapping networks $F_T$ and $F_S$ to map the features text description $T$ and spatial relation $S$ into a common latent space by reducing the reconstruction loss as follows:

$$L_{rec} = \|F_S(S) - F_T(T)\| \qquad (3)$$

where the $\|, \|$ is the $L_2$ distance function loss.

The model is trained by reducing the difference between attributes of the predicted object and the ground-truth object in the scene. For the object category, we adopt the cross-entropy loss as follows:

$$L_{cat} = -\sum y_i log \hat{y}_i \qquad (4)$$

while $\hat{y}_i$ is the predicted category of the object, and the $y_i$ is the ground-truth category.

As for the object's location, angle, and size, we follow [16] and model them with a mixture of logistics distributions. For the object attributes $[s, t, r]$, where object size $s \in R^3$, object location $t \in R^3$ and object rotation $r \in R^1$. We apply the mixture of logistic distributions to model them, take the object size $s$ as an example:

$$s \sim \sum_{k=1}^{K} \pi_k^s logistic(\mu_k^s, s_k^s) \qquad (5)$$

where $\pi_k^s$, $\mu_k^s$ and $s_k^s$ are the weight, mean and variance of the $k$-th logistic distribution. Thus, we can apply the loss to maximize the log-likelihood of training the model. During the test stage, we can sample the parameters of the objects from the predicted distributions.

## 5 THE NEW METRICS FOR TEXT-DRIVEN 3D SCENE GENERATION

To evaluate the quality of the generated scenes, current works[13, 17] have adopted the FID score, KID score, and CKL score to evaluate the generated scenes. The FID and KID scores measure the quality of the generated scenes, while the CKL score reflects the category distribution of the generated scenes. However, none of the metrics above can reflect the abilities of the approach to capture the semantics of the text description to form the 3D scenes. Thus, we propose new metrics from local and relation metrics to address the issues above. The local metric indicates the accuracy of the pair relation between the objects, while the relation metric evaluates the model's ability to address different relation types.

The local metric aims to evaluate the accuracy of the pair relation between the objects, which statistics the accuracy of the relations at the level of sentences.

**Definition 1: Local Accuracy**. The description sentence contains objects $A$ and $B$ with relation $R = r_1, r_2, .., r_n$. Then, the sentence accuracy is defined as follows:

$$SA(A, B, R) = \frac{max \sum_k 1 : r_k \in r(A^*, B^*)}{n} \quad (6)$$

where $A^*, B^*$ is the objects in the generated scene, which have the same category label with the $A, B$. $r(A^*, B^*)$ is the relation of the $A^*, B^*$ in the generated scene. $n$ is the amount of the relation in a sentence. Then, Local Accuracy (LA) for a scene $S$ is defined as follows:

$$LA(S) = \frac{\sum_i SA(A_i, B_i, R_i) * n_i}{\sum_i n_i} \quad (7)$$

where $A_i, B_i, R_i$ is the objects and relation of the $i$-th sentence in the text description. $n_i$ is the amount of the relation in the $i$-th sentence.

Local accuracy is the fundamental metric for evaluating each description pair-wise relationship. If there is a pair of objects in the generated scene that have the same object types and relationship, we think this relationship in the description has been accurately generated by the approach. We can adapt the **Mean Local Accuracy (MLA)** of all the scenes as the first metric in our benchmark.

We also hope to investigate the abilities of the approach to address different types of relationships. Thus, we design another metric in our benchmark, namely relation accuracy.

**Definition 2: relation accuracy**. Let $(A_i, B_i, R_i) \in TA$ be the set of all object-relation pairs of the description sentences mentioned in Definition 1. $(A_i^*, B_i^*, R_i^*) \in GA$ is the corresponding set of the selected object pairs in the generated scenes to maximize the $SA(A_i, B_i, R_i)$. Then, relation accuracy (RA) of relation type $R_T$ can be defined as follows:

$$RA(R_T) = \frac{\sum_i 1 : r_t \in R_i \wedge r_t \in R_i^*}{\sum_i 1 : r_t \in R_i} \quad (8)$$

where $r_t$ is the relation in the relation type $R_T$. Similarly, we can obtain the **Mean Relation Accuracy (MRA)** of all the types of relationships in the description as the second metric in our benchmark.

## 6 EXPERIMENT

In this section, we introduce the experiments of this paper and analyze the effectiveness of our proposed approach. First, we briefly introduce the dataset, the evaluation protocol, and implementation details. Then, we compare the proposed approach with the state-of-the-art methods in both quantitative and qualitative results. Furthermore, ablation studies are conducted to investigate the effectiveness of different components in our benchmark approach.

### 6.1 Datasets

The experiments are conducted on our proposed dataset RelScene, which is extended from the 3D-Front dataset[6] with the annotated text descriptions. For a fair comparison, we follow the current works[17][10] to adopt the same dataset filtering procedures of the ATISS and conduct experiments on three types of indoor rooms. The few-shot setting experiments are conducted in the Bedroom, which only utilizes the text description of the 5% scenes, while the rest of the text descriptions of the scenes are unavailable.

**Table 1: Performance comparison on Few-shot Text-conditioned 3D scene generation with template description**

| Method | MLA(↑) | MRA(↑) | FID(↓) | KID(↓) | CKL(↓) |
|---|---|---|---|---|---|
| ATISS* | 0.156 | 0.143 | 19.48 | 2.05 | 0.95 |
| w/o Psesudo | 0.224 | 0.204 | 18.75 | 1.95 | 0.92 |
| Our | **0.330** | **0.295** | **18.12** | **1.72** | **0.54** |

**Table 2: Performance comparison on Few-shot Text-conditioned 3D scene generation with natural language description**

| Method | MLA(↑) | MRA(↑) | FID(↓) | KID(↓) | CKL(↓) |
|---|---|---|---|---|---|
| ATISS* | 0.132 | 0.117 | 19.60 | 2.12 | 0.96 |
| w/o Psesudo | 0.159 | 0.132 | 19.12 | 1.97 | 0.90 |
| Our | **0.297** | **0.284** | **18.50** | **1.77** | **0.54** |

### 6.2 Evaluation Protocol

To evaluate the quality of the generated scenes, we first compare the approach with current works that are generated without semantic constraints. We first follow the current works[13, 17] focus on layout generation, which directly generates the scenes with location, angle, and the size of each object. To compare with those methods, we adopt the FID score, KID × 0.001 score, and CKL × 0.01 score in the experiments. For the FID and KID scores, we render the 256 × 256 top-down orthographic projections of generated and real scenes to extract the features for testing.

To evaluate the consistency of text, we also apply Mean Local Accuracy **(MLA)** and Mean Relation Accuracy **(MRA)** to evaluate the layout information predicted by the generation models, which are introduced in Section 5. We obtain the scene information of the objects, including the category, bounding box, and the angles in the scene, and calculate the relation of the objects in the generated scenes during the evaluation process.

### 6.3 Result analysis

*6.3.1 Few-shot text-conditioned scene generation.* Due to the annotated text description of the 3D scenes are not easy to obtain, we conducted the few-shot experiment to compare the abilities of the different models. We compare with the modified ATISS methods, which only utilize the scenes with the annotated text description for training. Similarly, we first compare our approach with the ATISS on MLA and MRA scores. Our approach improves the MLA scores from 0.156 to 0.330. We can observe that our approach has more significant improvement than text-conditioned scene generation in both two scores. This indicates that our approach fits the role of utilizing scenes without text descriptions to train the model under the few-shot setting. We can see that the performance of ATISS drops under the few-shot settings. It shows that the lack of annotated training data impacts the performance of the generation model. However, our approach proposes pseudo-feature generation to utilize the scenes without a description, which can alleviate the

**Table 3: Performance comparison on 3D scene generation without constraints**

| Metric | Method | Bedroom | Living | Dining |
|--------|--------|---------|--------|--------|
| FID(↓) | DepthGAN [24] | 40.15 | 81.13 | 88.10 |
|        | Sync2Gen [23] | 31.07 | 46.05 | 48.45 |
|        | ATISS [13] | 18.60 | 38.66 | 40.83 |
|        | DiffuScene [17] | 18.29 | 32.60 | 36.18 |
|        | Ours | **16.92** | **31.92** | **35.46** |
| KID(↓) | DepthGAN [24] | 18.54 | 50.63 | 63.81 |
|        | Sync2Gen [23] | 11.21 | 8.74 | 12.31 |
|        | ATISS [13] | 1.72 | 5.62 | 5.18 |
|        | DiffuScene [17] | 1.42 | 0.72 | 0.88 |
|        | Ours | **1.40** | **0.69** | **0.77** |
| CKL(↓) | DepthGAN [24] | 5.04 | 9.72 | 7.95 |
|        | Sync2Gen [23] | 2.24 | 4.96 | 7.52 |
|        | ATISS [13] | 0.78 | 0.64 | 0.69 |
|        | DiffuScene [17] | 0.35 | 0.22 | 0.21 |
|        | Ours | **0.33** | **0.20** | **0.20** |

impact of the lack of data, which only has a slight performance dropping compared. The pseudo-feature generation can sample pseudo-features based on the spatial relationships within 3D indoor scenes during the training process, which can effectively train the model with the scenes without description. As indicated in Table 1, the trends in method performance based on FID, KID, and CKL scores also demonstrate that our approach surpasses the ATISS.

*6.3.2 No semantic constraints scene generation methods.* Although our methodology tackles the generation of 3D indoor scenes driven by text, our approach can also compare with the methods that focus on scene generation without semantic constraints. As shown in Tab.3, our approach outperforms the existing approaches by achieving lower FID and KID scores across all four room types. These scores serve as metrics for evaluating the similarity between the generated scenes and real scenes. Compared to DiffuScene, we reduced the FID score from 18.29 to 16.92 in the bedroom, 32.60 to 31.92 in the living room, and 36.18 to 35.46 in the dining room. These lower FID and KID scores indicate that our approach produces scenes that closely resemble real scenes. Although we only apply the noise to replace the text description as input to generate scenes, the prior learned from the training stage can help to improve the quality of the scenes.

## 6.4 Ablation study

To investigate the effectiveness of our approach, we conducted the ablation study without pseudo-feature generation. The evaluation of MLA and MRA scores highlights the notable impact of pseudo-feature generation, especially in the few-shot setting. As evidenced

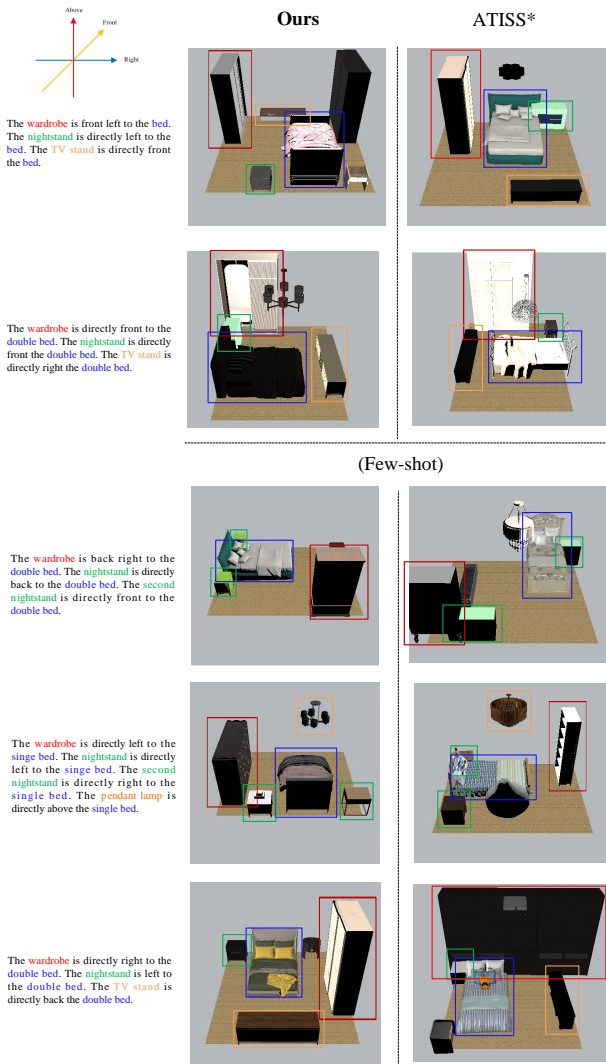

**Figure 3: Text-conditioned scene generation. Our approach can generate the scenes according to the description correctly.**

by the results of MLA and MRA scores, the approach incorporating pseudo-feature generation outperforms the one without it. This is attributed to the capability of pseudo-feature generation to efficiently sample pseudo-features based on spatial relationships, aiding model training with scenes lacking textual descriptions. Additionally, comparing FID, KID, and CKL scores reaffirms the efficacy of pseudo-feature generation.

We also compare the influence of the different amounts of annotated data on our proposed method. From Tab.4, we can observe that the method without pseudo feature generation has a larger fluctuation and lower performance than our proposed method. We can also observe that our approach has less improvement than the ablation method. It indicates that the pseudo-feature generation

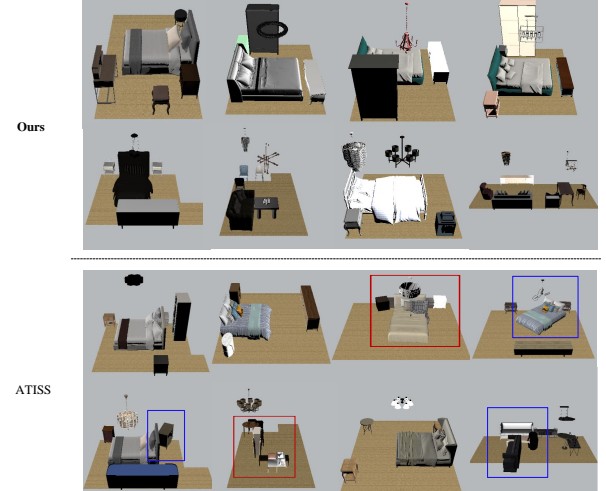

**Figure 4: Unconditioned scene generation. Red box: failure cases with objects overlapping. Blue box: failure cases of unreasonable combination of objects.**

**Table 4: Ablation study comparison on Few-shot Text-conditioned 3D scene generation with template description**

|         | method      | 5%    | 30%   | 50%   | 70%   |
|---------|-------------|-------|-------|-------|-------|
| MLA(↑)  | w/o Psesudo | 0.224 | 0.246 | 0.267 | 0.285 |
|         | Our         | **0.330** | **0.332** | **0.336** | **0.342** |
| MRA(↑)  | w/o Psesudo | 0.204 | 0.226 | 0.254 | 0.283 |
|         | Our         | **0.295** | **0.302** | **0.314** | **0.326** |
| FID(↓)  | w/o Psesudo | 18.75 | 18.62 | 18.42 | 17.95 |
|         | Our         | **18.12** | **18.05** | **17.92** | **17.85** |
| KID(↓)  | w/o Psesudo | 1.95  | 1.95  | 1.87  | 1.82  |
|         | Our         | **1.72** | **1.68** | **1.65** | **1.52** |
| CKL(↓)  | w/o Psesudo | 0.92  | 0.82  | 0.78  | 0.74  |
|         | Our         | **0.54** | **0.52** | **0.48** | **0.43** |

by our approach can approximately replace the text features in the training stage.

## 6.5 Visualization results of the approach

**Text-conditioned scene generation.** We also represent the visualization results of the text-conditioned scene generation in Fig. 3. The first two lines of results are trained with the full training set of the dataset. We can see that the ATISS fails to generate part of the spatial relation. For example, in the first line, the ATISS generates the "left" relation of the wardrobe and bed, which is close to the "front left" in the description. The position of the TV stand is incorrect and conflicts with the "front" relation in the description.

However, when we apply the few-shot setting for the model training, the ATISS almost cannot generate the scenes according to the description correctly. It is due to the training data is not enough for the model. However, our approach benefits from the pseudo-feature generation, which can effectively sample pseudo-features based on spatial relationships to train the model with the scenes without description. Thus, our approach can generate the scenes according to the description correctly.

**No semantic constraints scene generation.** Examining the visuals in Fig. 4, we compare the visualization results of our method to the original ATISS approach, focusing specifically on scenarios without text descriptions. Our method distinguishes itself by generating scenes with heightened diversity and improved plausibility, with fewer occurrences of object overlap. The illustration in Fig.4 distinctly illustrates the challenges faced by the ATISS method, particularly in dealing with significant issues of object overlap. For example, we used the red box to highlight the objects overlapping in Fig.4. The ATISS generates multiple objects without proper layout, leading to overlapping issues. We also use the blue box to highlight the unreasonable combination of multiple objects, such as inconsistent rotation of the bed(line 3, column 4), misaligned nightstand(line 4, column 1) and sofa(line 4, column 4). These challenges primarily arise from its insufficient ability to model spatial information about objects accurately during the autoregressive generation process.

## 7 CONCLUSION

This paper presents a text-driven 3D indoor scene generation method, which not only maintains consistent spatial relationships in alignment with the provided text description but can also be trained with a few language-annotated scenes. A new benchmark to evaluate the spatial relations in text-driven 3D scene generation. We extended the 3D-FRONT to construct a new dataset for text-driven 3D scene generation, which annotates the spatial relations of the objects in the scenes and provides two types of text descriptions, including template descriptions and nature language descriptions. Two new metrics are proposed to investigate the ability of the approach to generate correct spatial relationships among objects. The new metric offers a means to assess the precision with which the described spatial relationship is generated within the 3D scene. We conducted experiments on our newly proposed dataset to compare it with the current work with both new and traditional metrics, and the results show that our benchmark method showcases its superior flexibility in both scene qualities and spatial relations accuracy.

According to the results of the experiments, we can find that many issues and challenges should be addressed in future work. Through the experiment, the performance of the relation accuracy scores has great potential improvement, which indicates that the generation model should be further improved to address the whole scene. We can also observe the performance of the approach under the few-shot setting drops in all cases. Although our new proposed approach can decrease the impact of the lack of description, it is still a challenging task in further research.

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
