# OpenReview forum: "RelScene: A Benchmark and baseline for Spatial Relations in text-driven 3D Scene Generation"
_acmmm.org/ACMMM/2024/Conference — MM2024 Poster_

### Official Review · Reviewer_S5HZ · 2024-05-04

**Rating:** 5
**Confidence:** 4

**Summary:**

The paper focuses on text-driven 3D indoor scene generation, aiming to automatically create realistic rooms based on text descriptions. It introduces a new dataset and benchmark for assessing spatial relations in text-driven 3D scene generation, providing annotations for object spatial relations and text descriptions. The paper also proposes a pseudo description feature generation method for few-shot learning and new metrics to evaluate spatial relationships accuracy .

**Strengths:**

Introduction of a new dataset for text-driven 3D scene generation, addressing the lack of suitable datasets in this domain .
Development of a benchmark method to evaluate spatial relationships in text-driven 3D scene generation, enhancing research in this field .
Proposal of new metrics to assess the accuracy of spatial relationships in 3D scenes, contributing to the advancement of evaluation methods .
Conducting experiments on the proposed dataset to compare the benchmark method with current works, demonstrating superior flexibility in scene qualities and spatial relations accuracy .

**Limitations:**

The performance of the relation accuracy scores may require further improvement, indicating the need for enhancing the generation model to address the entire scene .
Challenges exist in maintaining performance under the few-shot setting, suggesting that overcoming the impact of the lack of description remains a challenging task for future research .

**Suitability:**

3

---

### Official Review · Reviewer_M7et · 2024-05-24

**Rating:** 2
**Confidence:** 3

**Summary:**

The objective of the paper is to construct 3D scene from textual description of placement of objects. The paper proposes a new dataset for text-driven 3D scene generation and an architecture to predict the placement of objects from a given scene layout and text description.

**Strengths:**

The paper proposes a new dataset for text-driven 3D scene generation and an architecture to predict the placement of objects from a given scene layout and text description. The paper also proposes new metrics to evaluate the quality of scene generated from the textual descriptions.

**Limitations:**

The major weakness I see for the paper is the practical need for an automatic system to create a 3D scene from the text description. With CAD models and scene layout available, the 3D scene creation from a textual description looks more like a force-fitted job for an AI system to perform. This can be rather performed simplistically using appropriate  packages that are in market  and the flexibility and accuracy they offer will be much more than textual description. Though the paper claims “textual description based method can boost productivity as compared to manual scene creation”, I tend to think the opposite. We do have most of the components ready for a manual scene creation   ( e.g. CAD models, scene layout, and the incomplete scene) . Hence the need for such a system is not justified.  The previous methods have tried to solve the room layout synthesis problem from a practical view point and have not used any textual description. They focused on semi-automatic methods where user interaction can get involved.

The paper needs to improve in  its writing quality. It seems sufficient care/time has not been given to writing. There are multiple places where writing related issues are there.
-	Figure 2 has ‘Angel’ in multiple places.
-	Table 4 has ‘Psesudo’ in multiple places.
Line 900 -  Incomplete line : “A new benchmark to evaluate the spatial relations in text-driven 3D scene generation.”
-There are no adequate caption descriptions of  Figure 1 & Figure 2

**Suitability:**

1

---

### Official Review · Reviewer_5UC9 · 2024-05-27

**Rating:** 4
**Confidence:** 3

**Summary:**

Text-driven 3D indoor scene generation is an important task to create realistic rooms with suitable objects and layouts. Addressing the issues from both the lack of consistent spatial relationships and the huge consuming of annotations, this paper built one database and presented a pseudo description feature generation method. Some experiments are conducted to verify the performance of the proposed method. Overall, sufficient and concrete work have been done in this paper. The presented method has limited novelty, and the comparison in this experiment is also limited.

**Strengths:**

(1) The paper conducts a dataset and benchmark for assessing spatial relations in text-driven 3D scene generation. This is meaningful.
(2) This paper provides a pseudo descriptive feature generation method for few-shot learning and proposes two metrics to evaluate the accuracy of spatial relationships.

**Limitations:**

(1) As said in Section 4.2, the attribute prediction network plays a crucial role in this paper. What is the detail illustration of this network and what are its advantages?
(2) In Figure 2, Object features are obtained after the processing of Object Encoder. Can we obtain the relationship between two objects based on the information like Location? In Line 3.2, this paper will generate natural language descriptions with ChatGPT. So whether this part should be illustrated in Figure 2?
(3) In Table 1 and 2, the comparison is mainly conducted with ATISS, one work published in 2021. It suggested more comparison and corresponding analysis in the paper.
(4) Readability needs to be improved. Some sentences are too long, which made hard understood.

**Suitability:**

3

---

### Meta-Review · Area_Chair_SCut · 2024-06-30

**Recommendation:** Accept (Poster)
**Confidence:** 5

**Metareview:**

This paper received one borderline accept, one weak reject and one weak accept final ratings from the reviewers. The weak reject reviewer main concern is that the text-driven 3D scene generation automatic system is not useful to boost generation productivity compared with routes using CAD models to design manually. However, text-driven 3D scene generation is an important task to create realistic indoor/outdoor scenes with suitable objects and layouts, which has been widely researched. AC agrees that this paper benefits from good writing and interesting idea. However, the authors are encouraged to make the necessary changes to the best of their ability.